# A Robust End-to-End Deep Learning-Based Approach for Effective and Reliable BTD Using MR Images

**DOI:** 10.3390/s22197575

**Published:** 2022-10-06

**Authors:** Naeem Ullah, Mohammad Sohail Khan, Javed Ali Khan, Ahyoung Choi, Muhammad Shahid Anwar

**Affiliations:** 1Department of Software Engineering, University of Engineering and Technology, Taxila 47050, Pakistan; 2Department of Computer Software Engineering, University of Engineering and Technology Mardan, Mardan 23200, Pakistan; 3Department of Software Engineering, University of Science and Technology Bannu, Bannu 28100, Pakistan; 4Department of AI, Software Gachon University, Seongnem-si 13120, Korea

**Keywords:** brain tumor detection, deep learning, MRI, TumorResNet

## Abstract

Detection of a brain tumor in the early stages is critical for clinical practice and survival rate. Brain tumors arise in multiple shapes, sizes, and features with various treatment options. Tumor detection manually is challenging, time-consuming, and prone to error. Magnetic resonance imaging (MRI) scans are mostly used for tumor detection due to their non-invasive properties and also avoid painful biopsy. MRI scanning of one patient’s brain generates many 3D images from multiple directions, making the manual detection of tumors very difficult, error-prone, and time-consuming. Therefore, there is a considerable need for autonomous diagnostics tools to detect brain tumors accurately. In this research, we have presented a novel TumorResnet deep learning (DL) model for brain detection, i.e., binary classification. The TumorResNet model employs 20 convolution layers with a leaky ReLU (LReLU) activation function for feature map activation to compute the most distinctive deep features. Finally, three fully connected classification layers are used to classify brain tumors MRI into normal and tumorous. The performance of the proposed TumorResNet architecture is evaluated on a standard Kaggle brain tumor MRI dataset for brain tumor detection (BTD), which contains brain tumor and normal MR images. The proposed model achieved a good accuracy of 99.33% for BTD. These experimental results, including the cross-dataset setting, validate the superiority of the TumorResNet model over the contemporary frameworks. This study offers an automated BTD method that aids in the early diagnosis of brain cancers. This procedure has a substantial impact on improving treatment options and patient survival.

## 1. Introduction

Cells in a biological being sometimes grow abnormally to form irregular volumes, which may affect the function of nearby healthy and normal cells. Human brain cells can also become tumors for multiple reasons, such as mutations or unrestrained cell division, disrupting the brain’s normal functionality and damaging healthy brain cells [1,2]. Brain tumors are considered one of the most life-threatening diseases, and thousands of lives are claimed annually. Early diagnosis holds the key to effective treatment and patient management. A radiologist normally uses brain Magnetic Resonance Images (MRI) to identify brain tumors manually [3,4]. The manual process is error-prone and time-consuming, even for the most expert radiologists. The different types, shapes, and sizes of tumors make the manual process even more challenging from a diagnosis point of view [5]. To overcome the challenges in accurate BTD and identification, a reliable and accurate automated system [6] is inevitable for the assistance of radiologists. This study specifically focuses accurate classification of MRI images into tumorous and normal MR images.

Radiologists utilize various medical imaging modalities to analyze brain images for tumor detection [7]. The non-invasive nature of MRI has become one of the most frequently used techniques to detect brain tumors. Automated BTD utilizing MR images has been frequently studied. Conventional machine learning (ML) approaches for the detection of brain tumors using MRI consist of preprocessing, feature extraction, selection, and classification [8]. Since Feature extraction and selection necessitates prior knowledge of the problem domain, it is the most critical stage of an effective automated BTD system [9]. However, these traditional ML approaches are time-consuming, used on limited data, and require tailored feature extraction techniques. On the contrary, modern DL models perform better in image classification or detection. The automatic feature extraction process at the dense level of DL models enables it to compute reliable discriminative feature maps on a significant amount of labeled data; thus, no handcrafted feature extraction technique is required. Due to this, several studies have employed DL models to detect and classify brain tumors.

Several limitations are associated with the existing automated approaches for BTD, including performance issues and manual identification of tumor regions for the ML model to classify the area as normal or affected. On the other hand, DL techniques can automatically extract features without manual intervention; however, the DL models proposed for BTD in the literature require large memory and high computation power. Furthermore, the performance assessment based on various evaluation metrics other than accuracy (such as precision, recall, specificity, and F-Score) is also important.

To cope with these limitations, in this paper, we proposed a novel TumorResNet DL-based model for BTD. The proposed model comprises several layers, such as convolution, LReLU, and batch normalization (BN). In contrast to earlier approaches, the proposed methodology does not include feature extraction and the selection or segmentation in the pre-processing stage [10,11], which needs prior feature extraction or segmentation of tumors from the MRI scans. The proposed model employs filter-based feature extraction, which can be useful in achieving high detection performance. The proposed model is capable of classifying images into different classes. The proposed framework uses a convolutional layer and LReLU activation function to extract the high-level features from the MR images. The main contributions of this study are:We designed and implemented a fully automatic end-to-end TumorResNet DL framework for BTD.The proposed method is robust to variations in intensity, size, shape, angle, and location of brain tumors in the images.Detailed experiments are performed on a standard Kaggle dataset with two classes (normal/tumor) to prove the superiority of our framework over the contemporary approaches for BTD.Moreover, we have also performed cross-validation over the “Brain MRI Images for BTD” dataset to show the applicability of the proposed method in real-world scenarios.

The remaining sections of the paper are arranged as follows: Section 2 provides the details about the literature review (or related work); Section 3 explains the proposed methodology. Section 4 describes the details of the experiments conducted for performance evaluation. Section 5 is about the discussion of our approach. Section 5 concludes our work.

## 2. Related Work

Currently, the detection of brain tumors has received a lot of attention. In literature, different techniques have been proposed for BTD. These techniques include the traditional ML approaches [8] and DL-based approaches [7,9]. The exploration of current methods for the detection of brain tumors is discussed in this section. The overview of existing approaches for BTD is also presented in Table 1.

Typical processes in conventional ML approaches for classification include preprocessing, feature extraction and selection, dimension reduction, and classification [12,13]. The ability to extract features typically depends on the expert’s expertise in the relevant field. Using conventional ML techniques in research is a difficult endeavor for a novice. The classification accuracy depends on the retrieved features, a fundamental stage in classical ML. There are two different forms of feature extraction.

**Table 1 sensors-22-07575-t001:** Overview of existing approaches for BTD.

Author	Method	Images Details of the Dataset	Advantages	Limitation
Woldeyohannes et al. [14]	Two-dimensional discrete wavelet transforms (2D-DWT) are used for feature extraction and SVM for classification.	160 normal and 240 tumorous MRI	Satisfactory results on a small dataset	Testing on imbalance dataset
Selvaraj et al. [15]	Statistical features (mean & variance), features from gray level cooccurrence matrices (entropy & contrast), and Least Squares SVM.	833 tumorous and 267 normal MRI	Both linear and non-linear kernels are used	Testing on imbalance dataset
Srilatha et al. [16]	LBP for feature extraction and SVM for classification.	58 normal MRI and 100 tumorous	Computationally efficient	Results are reported for a small dataset
Mishra et al. [17]	Graph Attention AutoEncoder-CNN	510 tumorous and 461 normal	Good generalization ability	Computationally complex
Rai et al. [18]	A novel Less Layered and less complex U-Net CNN	155 tumorous and 98 normal	The model is less complex and fast	Very low accuracy on uncropped images
Neeraja et al. [12]	CNN	155 tumorous and 98 normal	The model is lightweight and efficient	Low classification accuracy
Cinar et al. [19]	Resnet50	155 tumorous and 98 normal	The model is efficient with good generalization ability	Low classification accuracy
Kiraz et al. [20]	weighted KNN	300 tumorous and 300 normal	The model performs well on the combined images of two datasets	Performance is highly influenced by the location and size of brain tumors

First, there are low-level (global) features like texture and intensity features, first-order statistics like skewness, mean, and standard deviation, and second-order statistics like wavelet transform, shape, and Gabor features. However, traditional ML feature extraction techniques have some limitations. Firstly, it focuses only on low-level or high-level features. Secondly, it uses handcrafted features which require domain-specific experience and knowledge. It needs the efforts for manual extraction of features, which can decrease the effectiveness of the BTD system. Because handcrafted features demand strong domain information (i.e., the location or position of the tumor in an MR image); hence, it is not an easy task and is prone to human errors. The location and position of the tumor region in the MR image, together with its margin, texture, shape, and size, are related to the most important information and distinctive features of brain tumors. It highlights the urgent need for robust automation for BTD that incorporates high-level and low-level features without needing custom features. The DL-based approaches address these problems due to automatic feature extraction, which is more effective and robust for classification and detection purposes [21,22].

DL-based approaches provide fully automated end-to-end systems for brain tumor classification and detection. DL models use convolutional and pooling layers to learn and extract the features from the images. In [12], the authors used a novel CNN model comprised of four convolutional and two fully connected layers. A dataset consisting of two types of images, i.e., tumorous and normal MRI images, was used to assess the effectiveness of the proposed CNN. Data augmentation was used to increase the size of the training dataset. In [23], the authors used a convolution neural network (CNN) to extract hidden features from MRI images. Then kernel extreme learning machines (KELM) were used for classification based on these features. A dataset consisting of three types of brain tumor MRI images, i.e., meningioma, glioma, and pituitary, was used to assess the effectiveness of the CNN and KELM model ensemble. In [24], the authors introduced CNN based on an automatic solution for BTD and grouping using MRI images. In [19], the Resnet50 DL model was employed for BTD. The last five layers of the Resnet50 architecture were removed, and eight new layers were added. Furthermore, the performance of Resnet50, Alexnet, InceptionV3, Densenet201, and Googlenet models were also compared to find the model with the highest performance for BTD. In [25], the authors trained Faster R-CNN from scratch using MRI brain tumor images. Faster R-CNN combines the pre-trained AlexNet DL framework and region proposed network (RPN). The AlexNet model was taken as a base model for brain tumor MR image classification. RPN was given the AlexNet convolutional feature map as input. Fifty brain MRI images from a dataset were used to evaluate the framework. In [26], the authors used fine-tuned EfficientNet-B0 CNN base network to detect and classify brain tumor images effectively. The image enhancement and data augmentation methods are utilized to enhance the quality of the MRI scans and increase the number of training samples. In [27], the authors proposed a new brain tumor segmentation (Adaptive Fuzzy Deformable Fusion (AFDF)-based segmentation) and classification (Optimized CNN with Ensemble Classification) approach for brain tumor classification. In [28], the authors employed ML algorithms to classify the MRI scans of three freely accessible datasets and numerous pre-trained deep CNNs to obtain significant features from the MRI scans. The outcomes demonstrate that the SVM with radial basis function kernel beats other machine learning classifiers. The authors of [29] used Spectral Data Augmentation-based Deep Autoencoder, whereas the authors of [30] used the CNN model for brain tumor MRIs classification into normal and tumorous.

A review of prior research discovered that there are many issues with current automated techniques for detecting brain tumors. For the detection of brain tumors, some methods make use of the manually designated tumor regions, which prevents them from being totally automated. DL approaches are gaining attention because of their automatic feature extraction capabilities; however, they require large memory and high computation power. Moreover, the DL models normally provide lower results with small datasets, which is very common in the case of medical image datasets. Furthermore, brain tumor detection approaches cannot achieve high detection and classification performance. Additionally, it is difficult to identify and detect brain tumors because of the variability in size, form, intensities, and location of brain tumors. To address the challenges associated with BTD, we proposed an effective end-to-end DL model for BTD. Our proposed DL-based model increases the performance of BTD and classification by automatically extracting both low-level and high-level features for classification.

## 3. Methodology

### 3.1. Study Approach

We proposed a robust end-to-end DL-based TumorResNet model for the effective and efficient BTD using MR images. We accomplished a two-class classification (tumor and normal) for the automatic detection of brain tumors to help doctors quickly identify tumor patients. We provided MRI scans as input to the framework to execute the proposed methodology. Figure 1 depicts the proposed method’s abstract view, which comprises image resizing, dataset splitting, model development, training, and evaluation. We evaluated the proposed model on the Brain_Tumor_Detection_MRI (BTD-MRI) dataset [31]. To further assess the classification performance of the TumorResNet model, we classified brain MRIs of the Tumor Classification Data (TCD) dataset into benign and malignant using the same approach as described in Figure 1. Furthermore, a TumorResNet architecture with twenty-three convolution layers and three FC layers is designed to classify brain MR images (tumor/normal). The proposed method’s components are described in more detail below.

#### 3.1.1. Image Resizing

The dataset’s images (input images) are resized into 224 × 224 pixels to ensure uniformity, reduce the computational cost of our model, and speed up the processing.

#### 3.1.2. Dataset Partitioning

We used 80% of the MRI scans for training and 20% for model testing, dividing the dataset into training and testing sets.

#### 3.1.3. TumorResNet Architecture Details

The TumorResnet model follows the basic structure of CNN. The proposed TumorResnet model is primarily inspired by the philosophy of Resnets. We have not used any pre-trained existing model. We developed and proposed a novel model based on the basic concepts of Resnet18 [32]. The core difference between the proposed TumorResnet model and Resnet18 is that we used LReLU as an activation function. Moreover, the proposed model has 23 learned, 20 convolutional, and 3 FC layers, and the FC layers are followed by dropout layers. Table 2 highlight the key differences between the Resnet18 model and the proposed TumorResnet model.

In the proposed research work, we proposed a novel and robust end-2-end TumorResNet DL model for BTD. The proposed model employs a total of 67 layers, including convolution layers, activation functions, and normalizations for feature extractions and fully connected, dropout, Softmax, and classification layers to classify brain tumors. Our model has 23 learned layers: 20 convolutional layers and 3 FC layers, as shown in Table 3, making it deeper than normal CNN. The proposed model’s initial layer (pre-processing layer) is the image input layer (which accepts 224 × 224 input MRI scans for processing). A total of 67 layers comprise the architecture, including one input layer, one maximum pooling layer, 20 convolutional layers, 20 BN, 18 LreLU, three fully connected layers, two dropout layers, a Softmax layer, and a classification layer. To reduce the overfitting of the data, a dropout of 0.5% is applied to the fully connected layers. The first convolution layer performs 7 × 7 convolutions followed by BN, LreLU, and 2 × 2 max-pooling functions. These layers are followed by eight residual blocks (RB). The RB consists of two convolutional layers, followed by BN and LreLU activation functions; however, the second Relu activation is added after addition. Each RB is connected to the next RB through shortcut connections. The convolution layers perform 3 × 3 and 1 × 1 convolutions (downsampling layer). The stride parameter of both convolution layers, i.e., 3 × 3 convolution layer and 1 × 1 convolution, is set as 2. This layer is responsible for reductions in the input size. The small convolutional filters capture the most important features from the MR images. The features extracted using the RB are fed to the fully connected layers, and then Soft-max activation is performed to determine the classification probabilities. Table 3 provides information about the architecture of the proposed model. The residual block on TumorResNet is defined as follows
(1)y=F(x, W+x)
where *x* is the input layer, *y* is the output layer, and the *F* function is represented by the residual map.

As the dying ReLU issue progressively renders a significant percentage of the network inactive, it is undesired [33]. To tackle the dying ReLU problem, we employed the LreLU [34] activation function in the first full convolution layer. LreLU uses a tiny positive value, such as 0.1 rather than 0, to activate all neurons for most training instances. The LreLU function aids in expanding the ReLU function’s coverage area. LreLU assures that every neuron in the network should participate in the network’s active operation. Additionally, dropout layers are employed to minimize overfitting, and BN layers are used to decrease the training or learning time, particularly when working with huge datasets.

## 4. Experiment and Results

### 4.1. Hyper-Parameters

The performance of DL models tremendously depends on the choice and selection of hyper-parameters (learning rate, epoch size, kernel size, etc.), which are traditionally decided with a trial-and-error-based technique to identify the optimal value for each hyper-parameter. Table 4 lists the specifics of the chosen hyper-parameters. The proposed TumorResNet model was trained using the stochastic gradient descent method (SGD). The TumorResNet model is trained over 30 epochs to detect brain tumors while taking overfitting into account. In the identical experimental settings listed in Table 4 for the identification of brain tumors, our developed framework and other existing algorithms are trained and validated using the training and testing sets.

### 4.2. Dataset

This section includes comprehensive details regarding the dataset utilized for BTD. The research dataset utilized for detecting brain tumors in this study is adopted from the BTD-MRI dataset, which is easily accessible on Kaggle [31]. The dataset comprises two collections. The first collection of the dataset has 1500 MRI images without tumors, whereas the second collection has 1500 tumor images. We used 80% of the data for training and 20% (remaining) for testing the TumorResNet model. More specifically, we used all 3000 images of the standard Kaggle BTD-MRI dataset [28] for this experiment, where 2400 images (1200 tumorous and 1200 healthy images) were utilized for model training and the rest 600 images (300 tumorous and 300 healthy images) for testing. Some representative samples from the dataset are shown in Figure 2.

### 4.3. Evaluation Metrics

To assess the performance, we used the precision, accuracy, sensitivity, specificity, and *F*1_*score* metrics in this research. All the performance metrics are calculated as follows:(2)ccuracy=(TN+TP)/TS
(3)$$Precision=\;\frac{TP}{TP+FP}$$
(4)$$Sensitivity\;\left(recall\right)=\frac{TP}{TP+FN}$$
(5)$$Specificity=\frac{TN}{TN+FP}$$
(6)$$F1\_score=2\times \frac{Precision\times Recall}{Precision+Recall}$$
where *TN*, *FP*, *FN*, *TP*, and *TS* denote the true negative, false positive, false negative, true positive, and a total number of samples, respectively.

### 4.4. Experimental Setup

We performed all the experiments on a computer system equipped with Intel I CoITM) i5-5200U CPU and RAM of 8GB. For implementation, we used the R2020a version of MATLAB. The details of the experimental setup are mentioned in Table 5. For each experiment, the dataset is split into training and testing sets. The effectiveness of the proposed TumorResNet model for the BTD is evaluated through several tests. A detailed explanation of the experiments and their findings is provided in this section to elaborate on the effectiveness of the TumorResNet proposed approach.

#### 4.4.1. Performance Evaluation of BTD

This experiment aims to evaluate the BTD performance of our TumorResNet DL-based framework. We used all 3000 images of the standard Kaggle BTD-MRI dataset [31] for this experiment, where 2400 images (1200 tumorous and 1200 healthy images) were utilized for model training and the rest 600 images (300 tumorous and 300 healthy images) for testing. The proposed TumorResNet training framework went through 540 total iterations, averaging 18 iterations per epoch, throughout the 30 epochs. At epoch 30, the proposed TumorResNet framework achieved the highest classification accuracy, precision, recall, specificity, F1-score, and area under curve values of 99.33, 99.5, 99.5, 100, 99.5, and 0.9997, which proves the efficacy of our approach for BTD. We have trained the proposed TumorResNet model several times and the results of the TumorResNet model were consistent. Furthermore, we stored the model with the best results. We have provided accuracy and loss in Figure 3 to verify the proposed technique’s training and testing performance.

To assess the detection performance of the proposed TumorResNet framework, a confusion matrix is utilized to measure the performance and find the number of misclassified and correctly classified data. According to the confusion matrix (Table 6), the proposed TumorResNet framework correctly classified 300 tumor images in the testing phase. Furthermore, the TumorResNet framework correctly classified 296 normal MR while misclassifying the remaining 4 images.

Figure 4 represents the Receiver operating characteristic (ROC) curve of the proposed TumorResNet model, which expresses the BTD performance of the TumorResNet framework. To calculate the ROC, we utilized the perfcurve MATLAB function. The ROC applies threshold values on outputs in the range [0,1]. The TP Ratio and the FP Ratio are determined for each threshold. The ROC curve illustrates the TP to FP ratio, demonstrating the classification model’s sensitivity. For the classifiers, the area under the curve (AUC) is a key evaluation parameter, demonstrating the degree of distinction across categories. It determines how well the model distinguishes across classes. The AUC value close to 1 shows a high competence level, and the model will better differentiate between tumor patients and normal people. The TumorResNet reported an AUC value of 0.9997, which is visible.

#### 4.4.2. Ablation Study

The phrase “ablation research” is now increasingly employed in the domain of neural networks [35] to track the effectiveness of the proposed model by examining the results of changing specific elements. A thorough ablation study was carried out to show the effectiveness of the TumorResNet architecture. To assess how each component of the deep learning architecture contributes to the representation of the entire network, an ablation study involves removing or replacing a portion of the architecture. More precisely, one model branch is eliminated at a time, and TumorResNet model performance is assessed without that branch. This ablation study aims to assess the stability of the TumorReNet architecture, which is particularly important to assess how these components affect the system’s performance. Two experiments were conducted as part of an ablation analysis by altering different parts of the proposed TumorResNet framework. These tests confirmed the significance of the activation function (LReLU) and the impact of the FC and dropout layers on the effectiveness of the proposed model. In the first experiment, we introduced a global average pooling layer before the final FC layer and replaced the LReLU function with a Relu activation function in the feature extraction layers. We also eliminated the first two FC layers (along LReLU and dropout layers). The two FC layers were eliminated in the second trial (including the LReLU and dropout layers). Table 7 provides a summary of the ablated models’ performance. The results show that removing or changing any component of the TumorResNet model worsens the framework’s performance.

#### 4.4.3. Tumor Detection Comparison with Hybrid Approaches

The performance of the proposed tumorResNet DL framework is assessed in this section by employing a hybrid experiment for tumor detection. It is asserted that much better classification performance can be achieved by placing an SVM classifier at the top of the model rather than a conventional deep CNN [36]. Therefore, we developed a hybrid strategy where we first extracted deep features employing the six popular DL-based frameworks. Then we utilized these features for training SVM with the linear kernel as a decision function. C and Gamma hyperparameter values are set to 1.0 and 0.1, respectively, as these options produced the best results. We used DL-based classification techniques from Alexnet [37], Resnet18 [32], Squeeznet [38], Darknet19 [39], Shufflenet [40], and Mobilnetv2 [41] in the proposed research. The dataset is categorized into testing and training and testing sets for this experiment, i.e., 20% of the data is utilized for framework testing and 20% for framework training. More precisely, we used all the 3000 MRI images (1500 tumor and 1500 normal MRI images) of the dataset named BTD-MRI, where 600 images (300 tumorous and 300 healthy images) for model testing and the remaining 2400 images (1200 tumorous and 1200 healthy images) were used for model training. We trained these algorithms using the identical experimental setting (using the hyperparameter values as our model), as shown in Table 4. Different frameworks require input images of varying sizes like darknet19 requires 256 by 256 images while resnet18 takes 224 by 224. The dataset images are automatically resized using improved image data sources before entering the network for feature extraction. We employed activations on deeper levels (the last FC or global average pooling layer), such as the fc8 layer and the final layer (FC layer) of Alexnet, because these layers contain more high-level information than earlier layers. These layers integrate the global spatial positions of the input features after activation functions to create distinct features (i.e., Shufflenet gives 1000 features in total).

The classification outcomes using a hybIid approach (i.e., deep features and the SVM) are displayed in Table 8. The features of all twelve frameworks and the SVM method yielded less accurate results when compared to TumorResNet. The suggested TumorResNet technique outperforms the previous s six hybrid models, according to the experimental data, and achieves a tumor detection accuracy of 99.33 percent. Resnet18 had the lowest accuracy of 96.33 percent, while Squeezenet was the second-best model and performed with an accuracy of 99.17%. It is identified that all hybrid comparative models have accuracy levels of at least 95%. The suggested TumorResNet technique effectively extracts more distinctive features from the MRI scans, which explains why it was more successful at detecting and identifying brain tumors. We ensured the extraction of more robust and detailed characteristics by using tiny filters with 3 × 3 and 1 × 1 dimensions. Furthermore, the BN method in the suggested model provides regularization, reduces generalization error, and standardizes the inputs to a layer for each mini-batch.

#### 4.4.4. Performance Evaluation of TumerResNet Model on TCD Dataset

To further assess and evaluate the performance of our model and prove the generalizability power of the proposed TumorResnet framework, we performed this experiment to classify brain tumors into benign and malignant using the freely accessible Tumor Classification Data (TCD) on Kaggle [42]. This dataset includes MRI images of benign & malignant tumors and normal brain scans. The dataset consists of two collections, test and train, with benign, malignant, and normal subcollections. Only 350 images each from collections of benign and malignant tissue are used. For the experiment, we used all 700 images (350 normal and 350 malignant). We used 80% of data for training and 20% of data for testing and used the same experimental setup as mentioned in the table in Table 4 for this experiment. The training of the proposed framework took 71 min and 54 s to separate the malignant and benign tumors. TumorResNet framework correctly classified 70 malignant tumor images in the testing phase. According to the confusion matrix, as shown in Table 9, the TumorResNet model correctly classified all malignant MR images. Furthermore, the TumorResNet framework correctly classified 66 benign MRIs while misclassifying the remaining four images. Despite the limited data in the dataset, the proposed TumorResNet framework attained satisfactory precision, accuracy, recall, specificity, and F1-score of 100%, 97.14%, 94.59%, 100%, and 97.22%, respectively, which shows the effectiveness of the TumorResNet model for separating malignant tumors from the benign tumors. The TumorResNet model achieved the AUC value of 0.9873, close to 1, as shown in Figure 5, depicting a high competence level, meaning that the TumorResNet model better differentiates between benign and malignant tumors.

#### 4.4.5. Cross Dataset Validation

To validate the generalization capability of the proposed TumorResNet model for BTD over cross-dataset situations. This experiment’s key goal is to prove the proposed TumorResNet framework’s applicability to real-world situations. For this purpose, we used two datasets, i.e., BTD-MRI [31] and Brain MRI Images for BTD [43]. The Brain MRI Images for BTD are a standard Kaggle dataset (freely available). The dataset contains 155 tumors and 98 normal (without tumor) MRI images. We trained our model on all 3000 images of the “BTD-MRI” dataset and tested it over all 251 images of the “Brain MRI Images for BTD” dataset. Both datasets contain different images, i.e., images are diverse in terms of the images’ angles, shapes, sizes, and intensities of brain tumors. Moreover, the resolution and formats of images in the datasets are also diverse. Despite the training framework on the “BTD-MRI” dataset and testing on unseen samples of the “Brain MRI Images for BTD” dataset, the proposed TumorResNet technique attained reasonable precision, accuracy, recall, specificity, and F1-score of 99.00, 99.21, 99.5, 98.72, 99.22, and 0.9899 for the cross-dataset scenario proving the generalization power of the proposed framework for classification of brain tumor MRI images. Thus, we claim that the proposed framework can reliably be used for BTD using MRI images under diverse conditions effectively.

## 5. Discussion

In this study, we proposed a TumorResNet DL-based model that can more accurately (99.33) identify brain tumors from MRI than competing models. As shown in Figure 5, the model’s training and testing accuracy increase after each epoch while its training and testing loss rapidly decreases. The training of the proposed TumorResNet model took 540 min and 41 s for BTD. However, this time relies on the number of epochs and iterations per epoch. The proposed model is contrasted with hybrid methods (DL + SVM) and current cutting-edge models that may be found in the literature. We have validated the system using another common, freely obtainable Kaggle dataset, “TCD”, to analyze further the performance and generalizability of the proposed TumorResNet framework [42]. The proposed framework performs admirably and outperforms cutting-edge and hybrid methods. Previous research on BTD utilized an imbalance dataset [14,15,16,17]. We used the same amount of brain MRI scans from healthy and tumorous conditions to address this problem. Our research approach performs well since the proposed TumorResNet model uses the LReLU activation function rather than the ReLu activation function. We also addressed the dying ReLU problem using the LReLU activation function. The DL network will remain dormant in case of a dying Relu problem. We implemented the proposed TumorResNet technique to solve this problem using an LReLU. The LReLU activation mechanism permits a small (non-zero) gradient when the unit is inactive. As a result, it keeps learning instead of coming to a stop or hitting a brick wall. As a result, the LReLU activation function enhances the proposed TumorResNet model’s feature extraction capacity, improving its tumor detection performance. The skip connections technique employed in TumorResNet solves the vanishing gradient and degradation problems. It will skip any layer that has a negative impact on the architecture’s performance and allow an alternative shortcut channel for the gradient to flow through. The skip connection adds the output from the preceding layer to a succeeding layer; thus, learning does not degrade from the initial layers to the final layer. Moreover, these outcomes are attributable to the fact that our proposed approach can properly extract the most distinct, robust, and in-depth features to represent the brain tumor MRI image for precise and trustworthy classification. The first convolution layers extract features (low-level) like color and edges etc. In contrast, deeper layers are responsible for extracting high-level features like an abnormality in the MR images.

Brain tumor manual detection takes a lot of time and effort. Additionally, the MRI scans’ noise and varying contrast reduce the clarity of the images. Consequently, it became challenging for clinicians to examine the MRI directly. This study offers an automated BTD method that aids in the early diagnosis of brain cancers. This procedure has a substantial impact on improving treatment options and patient survival. The proposed method offers a reliable and effective means to identify brain tumors using MRI, assisting the brain doctor in making decisions quickly and precisely.

Furthermore, an experiment was designed to assess the BTD performance of the TumorResNet model compared to other state-of-the-art methods [29,30] for detecting brain tumors using MRI images. In this experiment, we compared our work with those approaches which used the same dataset (Brain_Tumor_Detection_MRI). However, this is not a direct comparison due to differences in data preprocessing, training and validation procedures, and processing power employed in their methodologies. Nayak et al. [29] used a deep autoencoder and spectral data augmentation to identify brain tumors. The morphological cropping procedure was used to downsize and decrease noise in the raw brain pictures in the first step. The data-space problem with feature reduction is then resolved using the discrete wavelet transform (DWT). The framework consists of seven hidden layers, which are used for encoding and decoding images. The encoder has three hidden layers that activate Dense, BN, and ReLU. To reduce generalization errors, provide regularization, and speed up the learning process, the authors used the BN operation. Finally, the dense layer was used to classify brain tumor MRIs into normal and tumorous [29]. The proposed approach outperformed the existing approaches with an accuracy of 97% and an AUC ROC score of 0.9946 (Table 10). Lamrani et al. [30] proposed a CNN to identify a brain tumor’s existence. The major goal of the authors was to use CNN as a machine learning tool for BTD. The CNN model was comprised of different layers, i.e., convolution layers, Maxpooling layers, a flatten layer, and dense layers [30]. The proposed approach achieved a remarkable accuracy of 96% and an AUC ROC score of 0.96 (Table 10). This experiment (comparative analysis) reveals the effectiveness of the proposed framework for BTD from MRI scans. The proposed approach achieved the highest overall accuracy of 99.33% and AUC of 0.9997, and we identified that the proposed framework performed comparatively well in terms of accuracy. It is significant to note that because of the end-to-end learning architecture used in the proposed TumorResNet method, there are no isolated processes for feature extraction, selection, or segmentation.

Although the proposed approach produced encouraging results, we identified several shortcomings and offered some suggestions for further study. The several forms of brain tumors, including meningiomas, pituitary tumors, and gliomas, cannot be classified using the proposed method. It is unknown from the proposed TumorResNet approach how well the system recognizes brain tumors when employing additional imaging modalities, such as computer tomography (CT scans). In the proposed approach, we continually divide image data into a training set (80%) and a test set (20%). On the other hand, alternative divides could lead to various outcomes. Although the proposed strategy performed significantly well on two publicly accessible datasets, this study also has the flaw that its results have not been confirmed in actual clinical studies. This statement also applies to the vast majority of the models examined in this study.

In the future, we’ll try to use a larger dataset to employ the proposed methodology to show how well the TumorResNet algorithm works. As we have just compared the performance of the proposed model with hybrid approaches (DL + SVM), in the future, we will compare the results of our model with other transfer learning-based approaches in which we will use the FC layer instead of SVM for classification. In the future, we are also interested in identifying the performance of the proposed model in classifying MRI tumor images into more fine-grained classifications, including meningiomas, pituitary tumors, gliomas, etc., by considering alternative research datasets. In the future, we plan to evaluate the generalizability of the proposed TumorResNet model in more Tumor datasets or other medical datasets containing CT scans, MRI, or chest radiographs so that it can be used in practice to detect various diseases including tuberculosis, breast cancer, lunopacity, etc. Furthermore, we want to use genuine clinical contexts to verify the proposed approach findings to assess the TumorResNet approach. As a result, we will be able to compare the effectiveness of our suggested framework to experimental methods directly. The employment of extra layers or other regularization methods to handle a tiny image collection using a CNN model is another future possibility. Furthermore, in the future, including meningiomas, we will compare the results of our model with other transfer learning-based approaches.

## 6. Conclusions

In medical image processing, BTD is one of the most significant, laborious, and time-consuming activities since manual (human-assisted) classification can lead to inaccurate prediction and analysis. The end-to-end TumorResNet DL framework for the reliable, accurate, and automated detection of brain tumors has been provided in this work. Furthermore, using publicly accessible datasets, we have verified the robustness of the presented approach. A real-time dataset with various tumor locations, sizes, shapes, and image intensities was used for the experimental study. The accuracy of 99.33% for BTD has demonstrated the superiority of our framework over existing techniques. Experimental results show that the proposed model outperforms the existing BTD. Our proposed method achieved good accuracy for BTD with less pre-processing (no separate feature extraction or feature selection) compared to other techniques. In the future, we intend to reduce further the system complexity, memory space requirements, and computational time taken by the execution of the model. The same method may be used in the future to detect and study various disorders in other regions of the body (liver, kidney, lungs, etc.) and classify the types of brain tumors like benign or malignant. Additionally, to further generalize the proposed approach in detecting other important medical diseases [44] together with the brain MRI, we aim to identify and capture the performance of the TumorResNet model by training and validating it on the identification of Covid-19 [45] from chest radiograph images [34], pest detection [46], other popular brain tumor types [47], predicting heart diseases [48,49], and mask detecting & removal [50,51] to generalize it further.

## Figures and Tables

**Figure 1 sensors-22-07575-f001:**
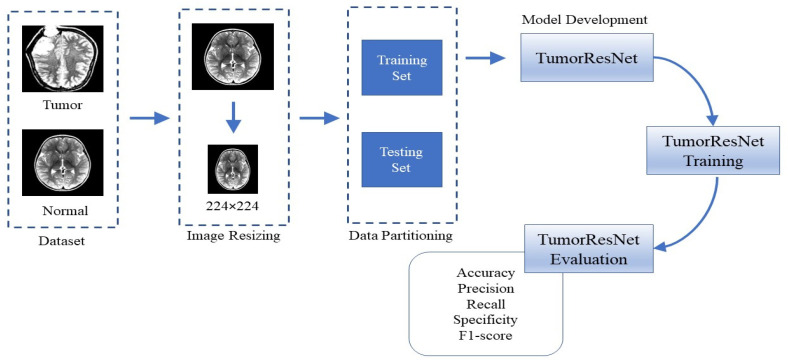
Flow diagram of the proposed approach for the BTD.

**Figure 2 sensors-22-07575-f002:**
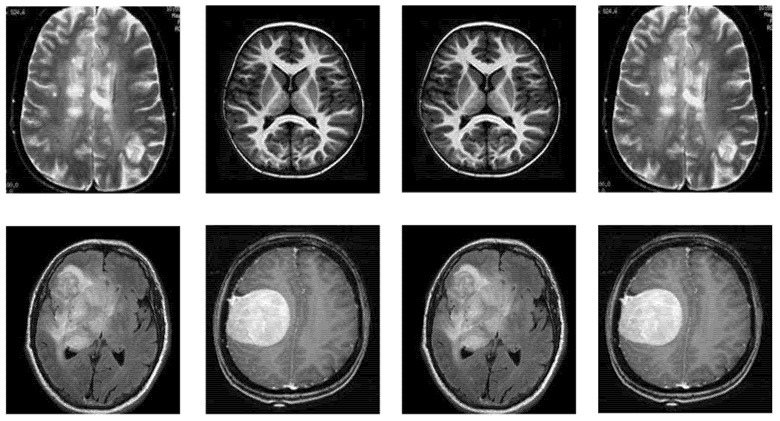
Samples of BTD-MRI dataset, upper row: No tumor examples and lower row: tumorous images examples.

**Figure 3 sensors-22-07575-f003:**
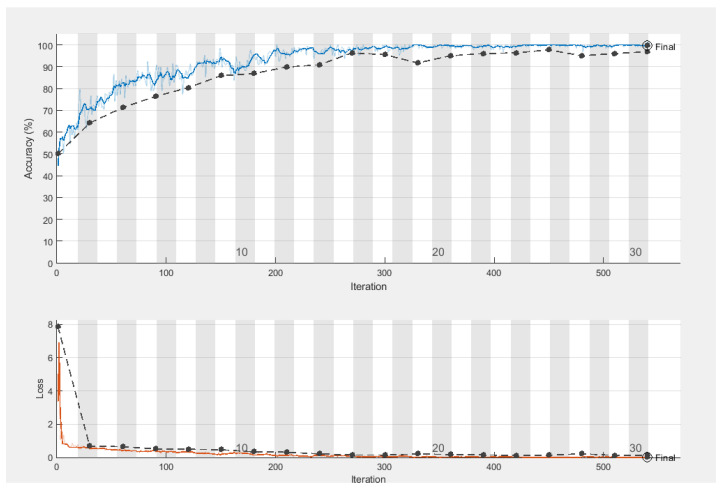
Training and testing accuracy of TumorResNet framework (blue line represents training accuracy, the black line represents testing accuracy, and the red line represents the training loss wheras the black line in the loss section represents testing loss).

**Figure 4 sensors-22-07575-f004:**
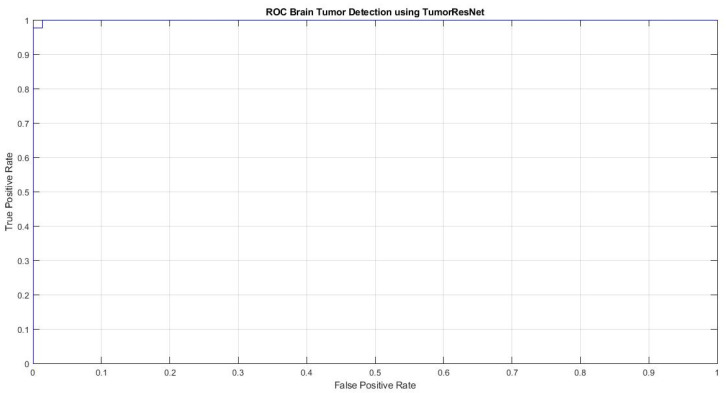
ROC plot of the proposed TumorResNet framework.

**Figure 5 sensors-22-07575-f005:**
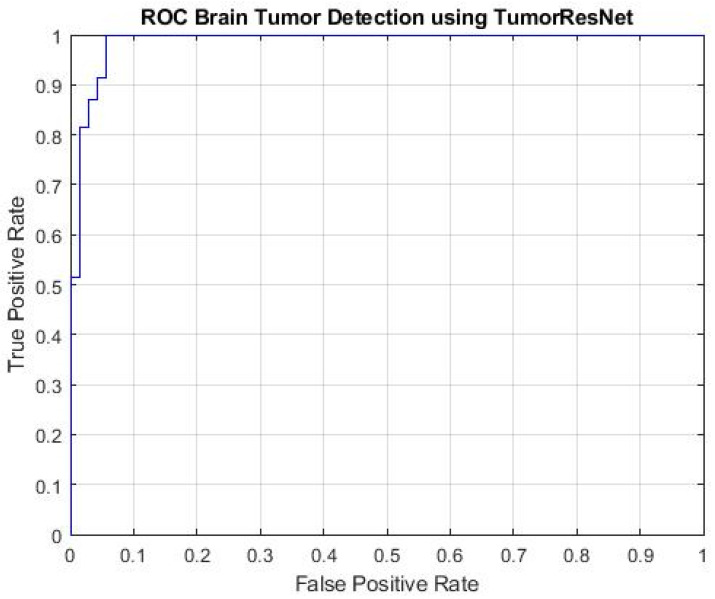
ROC plot of the proposed TumorResNet framework for benign and malignant BTD.

**Table 2 sensors-22-07575-t002:** Differences between the Resnet18 and TumorResnet model.

Property	Resnet18	TumorResnet
Total learnable layers	18	23
Total convolutional layer	17	20
Total fully connected layers	1	3
No of dropouts (0.5%)	0	2
No of global average pooling	1	0
Activation function	Relu	LreLU

**Table 3 sensors-22-07575-t003:** The TumorResNet Architecture.

S No	Layers	Filter	No of filters	Padding	Stride
1	Convolution (BN, LreLU)	7 × 7	64	3 × 3	2 × 2
2	Max-Pooling	3 × 3		1 × 1	2 × 2
3	Convolution (BN, LreLU)	3 × 3	64	1 × 1	
4	Convolution (BN, LreLU)	3 × 3	64	1 × 1	
5	Convolution (BN, LreLU)	3 × 3	64	1 × 1	
6	Convolution (BN, LreLU)	3 × 3	64	1 × 1	
7	Convolution (BN, LreLU)	3 × 3	128	1 × 1	2 × 2
8	Convolution (BN)	3 × 3	128	1 × 1	
9	Convolution (BN, LreLU)	1 × 1	128		2 × 2
10	Convolution (BN, LreLU)	3 × 3	128	1 × 1	
11	Convolution (BN, LreLU)	3 × 3	128	1 × 1	
12	Convolution (BN)	1 × 1	256		2 × 2
13	Convolution (BN, LreLU)	3 × 3	256	1 × 1	2 × 2
14	Convolution (BN, LreLU)	3 × 3	256	1 × 1	
15	Convolution (BN, LreLU)	3 × 3	256	1 × 1	
16	Convolution (BN, LreLU)	3 × 3	256	1 × 1	
17	Convolution (BN, LreLU)	3 × 3	512	1 × 1	2 × 2
18	Convolution (BN)	3 × 3	512	1 × 1	
19	Convolution (BN, LreLU)	1 × 1	512		2 × 2
20	Convolution (BN, LreLU)	3 × 3	512	1 × 1	
21	Convolution (BN)	3 × 3	512	1 × 1	
22	FC + LreLU + Dropout				
23	FC + LreLU + Dropout				
24	FC + Softmax + Classification				

**Table 4 sensors-22-07575-t004:** Parameters of the proposed architecture.

Parameter	Value
Optimization algorithm	SGD
Shuffle	Every epoch
Maximum Epochs	30
Iterations per epoch	18
Activation Function	LreLU
Validation frequency	30
Mini batch size	133
Verbose	false
learning rate	0.01
Dropout	0.5
Train Size	0.8
Test Size	0.2

**Table 5 sensors-22-07575-t005:** Details of the system used for implementation.

Sr. No	Name	Experiment Parameters
1	CPU	IntelI Core I i5-5200U
2	System type	Windows 10, 64 bit
3	Development tool	MATLAB R2020a
4	RAM	8 GB
5	ROM	500 GB

**Table 6 sensors-22-07575-t006:** Confusion matrix of the TumorResNet framework.

	Tumor	Normal
Tumor	300	0
Healthy or Normal	4	296

**Table 7 sensors-22-07575-t007:** Changing the network to evaluate the ablation study.

Experiment No	Activation Function	FC Layers	GAP	Dropout layers	Accuracy	Findings
Experiment 1	Relu	1	1	0	99.0	Accuracy dropped
experiment 2	LReLU	1	0	0	98.17	Accuracy dropped
Proposed method	LReLU	3	0	2	99.33	Best accuracy

**Table 8 sensors-22-07575-t008:** TumorResNet comparison with hybrid approaches.

DL Model	Accuracy	Precision	Recall	Specificity	F1-Score
Shufflenet	98.67	99	99	99.66	99
Mobilenetv2	98.33	98	98	99.32	98
Resnet18	96.33	96	96	97.28	96
Darknet19	98.67	98.5	98.5	98.34	98.5
Squeezenet	99.17	99.5	99.5	99.66	99.5
Alexnet	98.17	98.5	98.5	99.66	98.5
Proposed TumorResNet	99.33	99.5	99.5	100	99.5

**Table 9 sensors-22-07575-t009:** Confusion matrix of the TumorResNet framework for benign and malignant BTD.

Predicted Class			
		Tumor	Normal
Actual Class	Malignant	70	0
	Benign	4	66

**Table 10 sensors-22-07575-t010:** Comparison of the proposed work with existing methods.

Work	Method	Dataset	Accuracy	AUC	Date
Nayak et al. [29]	Spectral Data Augmentation-based Deep Autoencoder	BTD-MRI dataset	97%	0.9946	2022
Lamrani et al. [30]	CNN	BTD-MRI dataset	96%	0.96	2022
Proposed work	TumorResNet	BTD-MRI dataset	99.33	0.9997	2022

## Data Availability

A link to the research data will be provided upon request.

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
