# Peer review of "A Robust End-to-End Deep Learning-Based Approach for Effective and Reliable BTD Using MR Images"

_sensors, 2022, doi:10.3390/s22197575_

Round 1

Reviewer 1 Report

Authors proposed an interesting application based on Convolutional Neural Networks (CNN) for Brain Tumor Detection (BTD) from MR images. Although the results look promising, the article has a lot of flaws. The current version of the article is not at the level of JCR journals. The reviewer suggests that you consult several the more recent journal articles in this field as an example when rearranging and upgrading this article. For instance, you can use as an example the article:

Kang, J.; Ullah, Z.; Gwak, J. MRI-Based Brain Tumor Classification Using Ensemble of Deep Features and Machine Learning Classifiers. Sensors 2021, 21, 2222. https://doi.org/10.3390/s21062222

(The reviewer is not the author of this article neither is connected with authors in any way.)

There are many flaws in the article.

The Introduction is too long. It should focus on the problem that will be solved and what is proposed in the article. The motivation and contribution should be clear.

The Related work section should, first, cite some review articles from this research field. The old methods, based on hand-crafted features, should be just mentioned. The focus should lie on the newer Deep Learning methods (based on CNN). At the end of this section, advantages and shortcoming of state-of-the-art should be summarized.

Methodology is presented very unclear and flawed. The reader should be able to easily reproduce CNN topology based on descriptions, but this is not true for this article. Table 1 is also not sufficiently comprehensible and does not contribute to a better understanding. It is not clear from the descriptions what is your contribution in this CNN topology (novelty?). Did you just use one of the existing topologies? Hyperparameters and other settings belong at the beginning of Result section.

Result section. Clearly present used validation datasets (number of images: training/testing set; Is split predefined?). Newer papers are expected to verify their methods on multiple datasets, not just on one (as is case in this article)! Some datasets have more than two classes (classes with different tumor grades). Why did you not test your method on such datasets? In the Introduction, it should be explained the meaning of tumor grading (why, importance…). Experiments carried out in this work should be described in detail. The CNN training should be clearly presented (Weight initialisation? Image preprocessing? Augmentation? Early stopping? Any learning rate changing? Which model was stored: the last one or the best model on the validation set? Did you try to train the CNN several times? Were results consistent?). The time used for learning and prediction (inference) is not crucial for understanding the methodology, so it can be written down somewhere in the Discussion. How was ROC determined, because you don’t use any thresholds by classifications? Results section should contain just results, all explanations should be gathered in Discussion section (e.g., description about ReLU and Leaky ReLU…). In Section 4.3.2, besides SVM, why did you not use also FC as the classifier? Relatively old CNNs were used in this experiment. Section 4.3.3. State-of-the-art methods were not reviewed in the Related work section. The results in Table 6 are misleading, as they are obtained for some methods on different dataset as yours, and some metrics were even wrongly taken from the literature. (Review articles should be checked carefully as well.). Section 4.3.4. Cross dataset validation is not clear. Usually, you train the model on dataset 1 (all available training data should be used) and test on dataset 2, and vice versa.

Discussion section should be introduced.

References. There are missing information for some references (page number, year, volume, DOI link). For instance, [38] could not be found in IEEE Access.

English proofreading is mandatory.

Author Response

Dear Reviewer,

Thank you for allowing a resubmission of our manuscript, with an opportunity to address the interesting and beneficial comments and improve the overall quality of the paper by incorporating all the comments. Also, we highlighted the changes in the revised manuscript to make it visible and prominent.

We attached a point-by-point response to the comments, (b) an updated manuscript.

Reviewer 2 Report

Dear Authors,

After reviewing your manuscript, I found it interesting. The paper is well-written in my opinion and only some minor revisions are proposed. See the attached word document for details.

Kind regards,

Reviewer

Author Response

(The authors gave the same response as above.)

Reviewer 3 Report

In this article, the authors have proposed a robust end-to-end deep learning-based approach for effective and reliable brain tumor detection (BTD) using MR Images. Overall, this manuscript is well written but still requires many improvements and clarifications. A few of the shortcomings I am mentioning here.

1) In the related work, please include a comparison table that should highlight the strengths and weaknesses of the previous method and your proposed method. 

2) Please perform the experiment on at least one more dataset to detect tumors.

3) Please make a comparison table in the methodology section that should highlight the differences between the famous ResNet and Tumor ResNet model (proposed by you).

4) Please include the zoomed figure of the ROC curve instead of figure 4 which is almost blank. 

5) Please include the figure that should highlight some good and bad cases of detection by the proposed method. 

6) Please perform the ablation study to show the strength of your proposed method. 

Author Response

(The authors gave the same response as above.)

Round 2

Reviewer 1 Report

The authors have significantly upgraded and improved the article. A few minor ambiguities remain.

l. 241 – It should state ‘decrease’ instead of ‘increase’.

Unify denotation LReLU over article.

Ambiguity. Table 43 – 42 iterations per epoch, while in l. 291 – It is written 18 iterations per epoch. What is correct?

Table 4 – Missing batch size!

l. 486 – Rewrite the sentence as it is not a description of your method.

l. 491 - Rewrite the sentence as it is not a description of your method.

All used datasets in this study - It is unclear how many different patients were included in the study. Was the division of the data into training and test sets done according to the 'leave-patient-out' principle?  

Author Response

Dear Reviewer,

Thank you for allowing a resubmission of our manuscript, with an opportunity to address the interesting and beneficial comments and improve the overall quality of the paper by incorporating all the comments. Also, we highlighted the changes in the revised manuscript to make it visible and prominent.

We attached a point-by-point response to the comments and (b) an updated manuscript.

Reviewer 3 Report

Most of my comments are addressed. I recommend acceptance of this manuscript in its current form. 

Author Response

Dear Reviewer,

Thank you very much for accepting the paper in its current form. We appreciate your time in providing quality reviews to improve the overall quality of the research article.